

# Effect of salt stress and nitrogen supply on seed germination and early seedling growth of three coastal halophytes

Yanfeng Chen[1], Yan Liu[1], Lan Zhang[2], Lingwei Zhang[3], Nan Wu[4] and Huiliang Liu[2,5]

[1] School of Geography and Tourism, Qufu Normal University, Rizhao, Shandong, China
[2] State Key Laboratory of Desert and Oasis Ecology, Xinjiang Institute of Ecology and Geography, Chinese Academy of Sciences, Urumqi, Xinjiang, China
[3] College of Life Sciences, Xinjiang Agricultural University, Urümqi, Xinjiang, China
[4] School of Resources and Environmental Engineering, Ludong University, Yantai, Shandong, China
[5] Yili Botanical Garden, Xinjiang Institute of Ecology and Geography, Xinyuan, Xinjiang, China

Corresponding author
Huiliang Liu, lhl2033904@163.com

## ABSTRACT

Due to high salinity and low nutrient concentrations, the coastal zone is considered as one of the most vulnerable of the earth's habitats. Thus, the effect of salt and nitrogen on growth and development of coastal halophytes has been extensively investigated in recent years, but insufficient attention has been paid to the crucial stages of plant establishment, such as seed germination and seedling growth. Thus, we carried out a field experiment to evaluate the effects of salt stress (6, 10 and 20 g/kg NaCl) and nitrogen supply (0, 6 and 12 $gm^{-2}year^{-1}$) on seed germination and seedling growth of three coastal halophytes (including two dominant herb species *Glehnia littoralis* and *Calystegia soldanella*, one constructive shrub species *Vitex rotundifolia*) from September 2020 to June 2021. The results of our experiment showed that seeds of *G. littoralis* exhibited an explosive germination strategy in the early spring of 2021 with 70% of the seeds germinating. Conversely, the seeds of *V. rotundifolia* exhibited slow germination in the late spring of 2021 with only 60% of the seeds germinating. *C. soldanella* seed germination exhibited two obvious peak periods, but only 6% of the seeds germinated, which means that most seeds may be stored in the soil by stratification or died. All three halophytes showed greater sensitivity to nitrogen than salt stress during the seed germination stage. Nitrogen supply significantly delayed seed germination and reduced the cumulative germination percentage, particularly for *G. littoralis*. Despite the large impact of nitrogen on seed germination, nitrogen had a larger impact on seedling growth suggesting that the seedling growth stage of halophytes is more vulnerable to changes in nitrogen supply. Moreover, nitrogen supply significantly reduced the individual biomass of *G. littoralis*, *C. soldanella* and *V. rotundifolia*, with greater decreases seen in the dominant species than in the constructive species. Conversely, nitrogen supply increased underground biomass allocation of *G. littoralis* and *C. soldanella*, suggesting that the constructive species were less sensitive to nitrogen and exhibited a stronger anti-interference ability than the dominant species. Therefore, increasing nitrogen supply may firstly affect the seed germination and seedling growth of the dominant species, but not the constructive species.

# INTRODUCTION

The coastal zone is the transition zone between the ecosystems of the land and the sea, which means it has high biological diversity and is an ecosystem that has a large ecological and economic impact (*Ray, 1991*). The coastal zone also has a complex topography being shaped by the tides and is considered one of the most vulnerable of the earth's habitats (*Jickells, 1998*; *Yu et al., 2014*). Due to the periodic movement of the tides, the soil salt content often changes significantly in the coastal zone (*Yu et al., 2014*). These changes in soil salinity inevitably affect the whole process of plant growth and development, such as seed germination, seedling establishment, and reproduction growth. Only halophytes can adapt to the significant changes in salinity in the coastal zone (*Badreldin et al., 2015*). Furthermore, with the increase in both fertilizer use and the burning of fossil fuels, more and more nitrogen is being released into the coastal ecosystem through atmospheric nitrogen deposition and surface runoff, often resulting in the eutrophication of coastal water and soil (*Leeuw et al., 1999*; *Smith et al., 2015*; *Thibodeau, Bauch & Voss, 2017*). Therefore, it is of great significance to explore the response of coastal halophytes to salt stress and nitrogen supply in coastal habitats for population continuity and community stability.

Halophytes are unique plants that complete their life cycle in high salt habitats, and are better able to adapt to salt stress than non-halophytes through salt avoidance, evacuation, and accumulation of toxic ions (*Kosová et al., 2013*; *Flowers, Munns & Colmer, 2015*). Meanwhile, numerous studies have shown that a certain concentration of salinity often promotes the growth of halophytes, but excessive salt stress can cause these plants to lose water, inhibits plant growth, and even causes plant death (*Li, Ma & Li, 2007*; *Ali & Yun, 2017*). For example, salt concentrations of 250, 750 and 2,250 ppm significantly promoted the germination and seedling development of *Lolium multiflorum*, while a salt concentration of 10,000 ppm led to a germination percentage of only 7% and seedling development stopped completely (*Shao et al., 2020*). Halophytes also exhibit species-specific responses to salt stress. For example, *Tobe, Li & Omasa (2000)* analyzed the salt tolerance of *Haloxylon ammodendron* and *Haloxylon persicum* in the seed germination and seedling stages and found that the salt tolerance of *H. ammodendron* during the rapid growth stage is higher than that of *H. persicum*, and that this difference in salt tolerance explains the difference in the geographical distribution of these two species. Moreover, the adaptability of halophytes to saline conditions can also differ between the seedling stage and reproduction stage. For example, adult plants have developed root systems and salt-tolerant physiological mechanisms, so they have strong salt stress resistance while seedlings often lack a complete tolerance mechanism and are exposed to the large environmental fluctuations at the surface of the soil, resulting in higher mortality (*Yohannes et al., 2020*). Therefore, the response of halophyte to salt stress is affected by salt concentration, species specificity, and growth stages of the halophyte.

Nitrogen is a basic element of organic molecules, but it is also the main limiting nutrient for plant growth in most terrestrial ecosystems (*Santi, Bogusz & Franche, 2013*; *Hestrin et al., 2021*). Thus, many researchers have investigated the effects of nitrogen supply on crop yields and grassland productivity. For example, in agro-ecosystems, nitrogen supply has long been considered the main environmental factor limiting crop yield (*Möller et al., 2008*). In the northern Loess Plateau of China, nitrogen supply significantly improved the community productivity and induced changes in the community composition and species diversity of herbaceous plants (*He, Liu & Xie, 2015*). Conversely, due to the extremely poor nutrients supply in coastal zones, the effects of nitrogen supply on the growth of halophytes have not attracted the same scholarly attention (*Shiozaki et al., 2015*). The soil nitrogen content in the coastal zones is increasing as nitrogen-producing activities increase, and these changes in soil nitrogen inevitably affect the growth of coastal halophytes (*Prescott, Chappell & Vesterdal, 2000*; *Mccrackin, Harrison & Compton, 2014*). For example, nitrogen addition not only significantly improved the nutrient status of halophytes, but also enhanced the salt tolerance and osmotic adjustment in plants, thereby alleviating plant harm from salt content (*Pataki et al., 2008*; *Hamed et al., 2013*). Nitrogen supply can also correct underlying imbalances in plants exposed to salinity, but for some plants with weak salt tolerance, increased nitrogen fertilizer under high salt stress results in a significant decrease in nutrient content and individual biomass (*Song et al., 2010*; *Song & Xing, 2010*). Therefore, the effects of nitrogen supply on the growth of halophytes may be affected by the salt tolerance of plants, and further studies on the interaction between salt and nitrogen are urgently needed.

In the coastal zone of Shandong Peninsula, northern China, *Glehnia littoralis, Calystegia soldanella* and *Vitex rotundifolia* are three typical halophytes that grow on beaches not far from the coast (*Bian, Li & Zhang, 2007*). Thses plants also showed significant tolerance to salt and barren soil, and have been regarded as pioneer species for windbreak, sand fixation, soil improvement, and beach greening (*Zhao & Fang, 2005*; *Bian, Li & Zhang, 2007*). In recent years, due to both natural and human factors, soil salt and nitrogen content have changed (*Mariotti et al., 2010*; *Mas-Pla et al., 2013*). These periodic changes in salt and nitrogen content in the soil affect the germination, survival and growth of halophytes. For most species, the seed stage is considered to be the most tolerant plant growth period of environmental stress, while the seedling stage is considered to be the most vulnerable period (*Muller-Landau, 2010*; *Milla & Lopez, 2014*). Thus, we hypothesized that the response of three halophytes (*G. littoralis, C. soldanella* and *V. rotundifolia*) to salt stress and nitrogen supply in the seedling stage would be larger than during the seed stage. *V. rotundifolia* has long been regarded as the constructive species in the coastal zone of Rizhao, and *G. littoralis* and *C. soldanella* are considered the dominant species of the community (*Bian, Li & Zhang, 2007*). The constructive species can often absorb more water and nutrients than the dominant species (*Yang et al., 2019*). Thus, our second hypothesis was that the response of constructive species (*V. rotundifolia*) would be more sensitive than the dominant species (*G. littoralis* and *C. soldanella*) in both the seed germination and seedling growth stages. Finally, we carried out salt stress and nitrogen supply experiments in order to compare the response of different halophytes to salt stress
and nitrogen supply in both seed germination and early growth. The adaptive strategy of the halophytes in the seed germination and seedling growth stages to salt stress and nitrogen supply are likely to provide a theoretical foundation and practice guidance for vegetation restoration and environmental improvement in the coastal zones.

# MATERIALS AND METHODS

## Study area

Rizhao City is located in the southeast of the Shandong Peninsula and covers an area of about 5,358 km$^2$ with a population of 2.96 million (*Song et al., 2017*). The Rizhao coast is a sand coast, and saline-alkali soil is widely distributed in the coastal zone of Rizhao (*Song et al., 2017*). Rizhao also lies in the north temperate zone with a temperate monsoon climate, hot and rainy in summer, and cold and dry in the winter (*Chen et al., 2012*). The average annual precipitation reaches 897 mm, and the average annual temperature is about 12.7 °C (*Chen et al., 2012*). The soil and climatic conditions of the study area have bred rich and unique halophytes in the coastal zones, such as *Salicornia europaea*, *V. rotundifolia*, *G. littoralis*, *C. soldanella*, *Apocynum venetum* (*Bian, Li & Zhang, 2007*). Halophytic vegetation is widely distributed in the coastal saline soil, and is important for wind mitigation and sand stabilization; some halophytes also have high medicinal value (*Zhao & Fang, 2005*; *Bian, Li & Zhang, 2007*; *Tamura, Kubo & Ohsako, 2021*). Therefore, studying the effects of salt stress and nitrogen supply on halophytes is of great significance for improving the ecological environment and economic development of the coastal zones.

## Community investigation and seed collection

To understand the performance of each species (*G. littoralis*, *C. soldanella* and *V. rotundifolia*) in a field habitat, we conducted a community investigation by selecting typical plots in the Wanpingkou Scenic Spot in Rizhao City and set three quadrats (3 × 3 m) in each plot from May to June 2020. We then investigated the species composition and number in each quadrant and measured the height and coverage of each species.
The importance values of each species were then calculated according to the results of our investigation (including height, coverage, and occurrence).

From June to October 2020, mature seeds of *G. littoralis*, *C. soldanella* and *V. rotundifolia* were collected from the field habitat (Fig. 1). Seeds of *C. soldanella* and *V. rotundifolia* were extracted from the inflorescence using sieves with different mesh sizes, but the seeds of *G. littoralis* could be directly obtained. The bulk seeds of each species were obtained from at least 50 individual plants. After manual cleaning, the dry seeds were brought back to the laboratory and stored in a 4 °C refrigerator. Until mid-October 2020, we selected the robust, full, and intact seeds of three halophytes for pretreatment using the experimental method of seed pretreatment by *Sheidai et al. (2012)*. Firstly, we disinfected them with 70% ethanol for 5 min and then poured out the alcohol. Secondly, we added a 6% sodium hypochlorite solution and mixed it by shaking for 15–20 min. Finally, we sowed the sterilized seeds in flowerpots filled with sand soil in test plots at the Rizhao Campus of Qufu Normal University (the test plots are only 5 km away from the Wanpingkou Scenic Area, and the temperature, precipitation and soil are the same as the

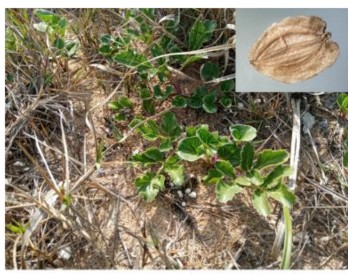
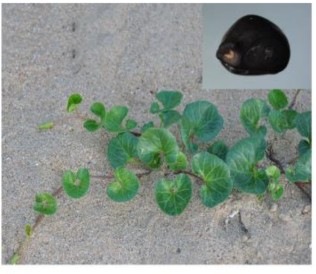
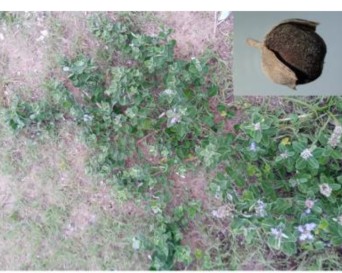

*Glehnia littoralis*          *Calystegia soldanella*          *Vitex rotundifolia*

**Figure 1 Plant and seed morphology of *Glehnia littoralis*, *Calystegia soldanella* and *Vitex rotundifolia in situ* soil.** Photographs taken by the first author in June 2020.

habitat condition of the three halophytes). The planting depth of seeds was about 2 cm deep from the surface of the sandy soil, and 100 seeds were dispersed evenly in each flowerpot. Each flowerpot was 30 cm in diameter, 32 cm tall, and filled with 10 kg of sand soil. The physicochemical properties of coastal sandy soil in the Shandong Peninsula showed that the soil salt content, total nitrogen, total carbon, $NH_4$-N and $NO_3$-N of the 10 cm soils in the Yellow River Delta were $6.01 \pm 0.53$ g/kg$^{-1}$, $0.55 \pm 0.08$ g/kg$^{-1}$, $15.7 \pm 0.21$ g/kg$^{-1}$, $119.3 \pm 11.2$ mg/kg$^{-1}$ and $3.45 \pm 0.50$ mg/kg$^{-1}$, respectively (*Guan et al., 2019*).

## Experimental design

This study included two experiments: a seed germination experiment and a seedling growth experiment. In both experiments, salt stress and nitrogen supply were tested against a control. The control treatment was based on the salt and nitrogen concentration in sand soil with no additional salt and nitrogen added (CK); $Na^+$ concentration *in situ* soil is about 5 g/kg (Fig. S1). The salt stress treatments were divided into two levels based on the variation range of soil salinity in different coastal dunes of the Shandong Peninsula (Fig. S1; *Song et al., 2018*): moderate salt stress (Na1, $Na^+$ concentration of about 10 g/kg) and high salt stress (Na2, $Na^+$ concentration of about 20 g/kg). Due to rapid agricultural and industrial development in China, total nitrogen deposition fluxes (dry and bulk) of the North China Plain have reached 6 gm$^{-1}$yr$^{-1}$ (*Fu et al., 2019*), and according to *Galloway et al. (2008)*, global N deposition will double by 2050 compared with the early 1990s. Thus, the selection of nitrogen level was based on the current nitrogen deposition rate and the potential amount that may occur in the future: moderate, or current nitrogen supply (N1, 6 gm$^{-2}$year$^{-1}$) and high nitrogen supply (N2, 12 gm$^{-2}$year$^{-1}$). Five treatments (CK, Na1, Na2, N1 and N2) were set for each species in the seed germination and seedling growth experiments, and each treatment consisted of eight replicates.

In the seed germination experiment, to avoid osmotic shock, NaCl was applied stepwise from 2 g/kg to reach the desired concentration for Na1 treatment: for Na2 treatment, 4 g/kg NaCl was applied stepwise to reach the desired concentration. Both groups reached final salt levels, either moderate salt stress or high salt stress, after 5 days. All flowerpots were fertilized beginning in October 2020 (seed sowing period) with granular slow-release ammonium nitrate ($NH_4NO_3$). $NH_4NO_3$ was first dissolved in pure water and then

sprayed evenly on the flowerpots. To avoid the effect of nitrogen shock, nitrogen was applied gradually by adding 1 g/kg per day for N1 treatment, and nitrogen was applied gradually by 2 g/kg per day for N2 treatment. Both groups reached final nitrogen levels, either moderate nitrogen supply or heavy nitrogen supply levels after 6 days. The same amount of pure water without nitrogen and NaCl was sprayed on the control flowerpots. To prevent salt and nitrogen accumulation, flowerpots were washed with distilled water after every four irrigation cycles. In the seedling growth experiment, we sowed *G. littoralis*, *C. soldanella* and *V. rotundifolia* seeds in flowerpots, but salt stress and nitrogen supply treatments were not applied before seedling emergence. In the spring of the second year, 2 weeks after seedling emergence, seedlings of *G. littoralis*, *C. soldanella* and *V. rotundifolia* of uniform heights and with the same number of leaves were selected for the seedling growth experiment. The experiment was done with a randomized design with salt stress and nitrogen supply treatments with eight flowerpots for each treatment and five plants in each flowerpot. The method of salt stress and nitrogen application was similar to the method used in the seed germination experiment.

## Seed germination and early seedling growth

For the seed germination experiment, we sowed seeds and recorded the number of seeding that emerged per day under salt stress and nitrogen supply treatments until no new seedlings were observed for 30 consecutive days. Finally, we calculated the main germination parameters, including the final germination percentage and cumulative germination percentage. The final germination percentage (%) = (Number of germinated seeds/total number of tested seeds) × 100%; the cumulative germination percentage (%) = (the cumulative number of germinated seeds during the experiment cycle/total number of tested seeds) × 100%.

In the seedling growth stage, 100 days after seed germination (approximately mid-June 2021), 10 seedlings were selected randomly for salt and nitrogen treatments, the plant height was measured with a ruler, and the number of leaves and branches was recorded. After measuring the above-ground traits, we carefully took out the whole plant using a shovel, and rinsed the roots with pure water and absorbed the water on the surface of the plant with absorbent paper in order to determine the root length and organ biomass. To determine the dry biomass of plant samples, we put fresh plant samples into envelopes and then put them in an oven at 105 °C for half an hour. We then baked them in an oven at 70 °C for 72 h to a constant weight, and weight the above-ground and under-ground biomass using a Sartorius BS210S electronic balance (0.0001 g). Finally, we calculated individual biomass (under-ground biomass + above-ground biomass) and root-shoot ratio (under-ground biomass/above-ground biomass).

## Statistical analyses

The seed germination, seedlings morphological traits and individual biomass were expressed as mean + standard error, and the arcsine transformation of seed germination percentage and cumulative germination percentage was performed before one-way analysis of variance. One-way analysis of variance (ANOVA, $p \leq 0.05$) was used to

**Table 1 Characteristics of plant community in the seed collection sites.**

| Family | Species | Life cycle | Height (cm) | Coverage (%) | Occurrence (%) | The important value |
|---|---|---|---|---|---|---|
| Convolvulceae | *Calystegia soldanella* | Herb | 6.00 | 7.50 | 50.00 | 0.20 |
| Apiaceae | *Glehnia littoralis* | Herb | 23.00 | 15.00 | 70.00 | 0.29 |
| Lamiaceae | *Vitex rotundifolia* | Shrub | 54.00 | 66.50 | 100.00 | 0.56 |
| Cyperaceae | *Carex kobomugi* | Herb | 10.00 | 1.00 | 20.00 | 0.08 |
| Cyperaceae | *Carex jiaodongensis* | Herb | 8.00 | 7.50 | 80.00 | 0.18 |
| Gramineae | *Avena fatua* | Herb | 16.00 | 2.50 | 20.00 | 0.11 |

compare the effects of salt stress and nitrogen supply on seed germination and seedling growth of three halophyte species, respectively. If ANOVA exhibited significant effects, Tukey's test was used to determine differences among salt stress and nitrogen supply treatments. Data was graphed using Origin 2019 (Systat Software, London, UK), and all statistical analyses were carried out using the statistical software of SPSS19.0.

# RESULTS

## Community investigation in field habitat

Based on the results of the investigation in the field, we found that *V. rotundifolia* (0.56) showed the largest importance in the field habitat, followed by *G. littoralis* (0.29) and *C. soldanella* (0.20), but *Carex jiaodongensis* (0.18), *Avena fatua* (0.11) and *Carex kobomugi* (0.08) showed the relatively small important values (Table 1). Thus, *V. rotundifolia* can be defined as the constructive species of the community in the coastal zone of Rizhao, but *G. littoralis* and *C. soldanella* can be regarded as the dominant species of the community.

## The effects of salt stress and nitrogen supply on seed germination

After sowing on 10 September 2020, *G. littoralis* seeds exhibited explosive germination in the early spring of 2021 with the seed germination percentage reaching more than 70% in just 1 week (Fig. 2). Conversely, *V. rotundifolia* seeds exhibited a slow germination in the late spring of 2021 with the seed germination percentage reaching about 60% after around 1 month (Fig. 2). For *C. soldanella*, seed germination exhibited two peak periods of germination. The first peak period of seed germination occurred in late autumn of 2020 with the germination percentage only reaching about 2%, but no seedlings overwintered successfully. The second peak period of seed germination occurred in the spring of 2021, but the spring germination percentage only reached about 6% (Fig. 2).

Salt stress has no significant effect on the seed germination of *G. littoralis* ($p = 0.14$, Table 2), but nitrogen supply significantly inhibited the seed germination of *G. littoralis* with N2 treatment inhibiting nearly all seed germination ($p < 0.05$, Table 2 and Fig. 3). Nitrogen supply significantly promoted the seed germination of *V. rotundifolia* ($p < 0.05$, Table 2 and Fig. 3), but the effects of salt stress and nitrogen supply did not significant impact the seed germination of *C. soldanella* ($p = 0.57$ and $p = 0.10$, Table 2 and Fig. 3).

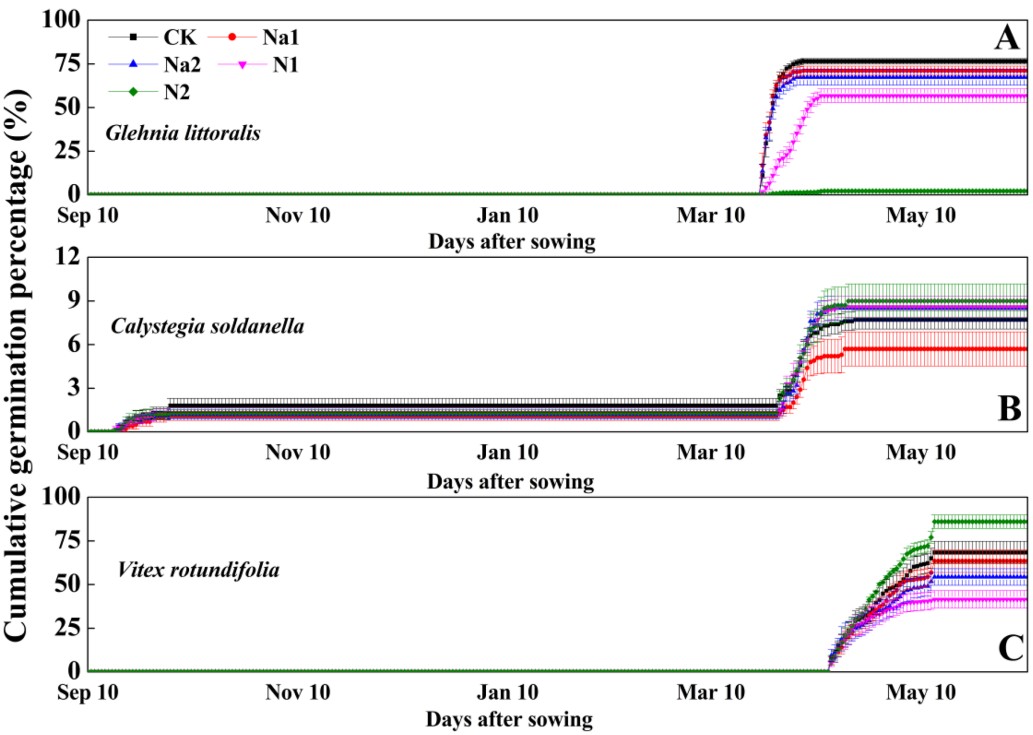

**Figure 2** Effects of salt stress and nitrogen supply on cumulative germination percentage of *Glehnia littoralis* (A), *Calystegia soldanella* (B) and *Vitex rotundifolia* (C). CK, no additional salt and nitrogen were added, Na1, Na$^+$concentration is about 10 g/kg, Na2, Na$^+$concentration is about 20 g/kg, N1, 6 gm$^{-2}$year$^{-1}$, N2, 12 gm$^{-2}$year$^{-1}$. Different lowercase letters indicate significant differences ($p < 0.05$) among CK, N1, N2, Na1 and Na2.                     

## The effects of salt stress and nitrogen supply on the early seedling growth

The height, root length and leaf number of *V. rotundifolia* were significantly larger than that of *G. littoralis* and *C. soldanella* ($p < 0.05$, Table 2). Nitrogen supply significantly inhibited the seedling growth of *G. littoralis*, *C. soldanella* and *V. rotundifolia*, including the height, root length and leaf number ($p < 0.05$, Table 2 and Figs. 4–6), and the higher of nitrogen concentration, the stronger the impact (N2 > N1). Conversely, salt stress had no significantly effect on seedlings growth, except Na2 inhibited the root length of *C. soldanella* and *V. rotundifolia* ($p < 0.05$, Table 2 and Figs. 4–6).

The individual biomass of *V. rotundifolia* was significantly larger than that of *C. soldanella* (0.267 g/plant) and *G. littoralis* (0.113 g/plant; Table 2 and Fig. 7). Nitrogen supply significantly reduced both the above-ground and under-ground biomass of *G. littoralis*, *C. soldanella* and *V. rotundifolia* ($p < 0.05$, Table 2). As for salt stress, Na1 has almost no effect on the biomass accumulation of *G. littoralis*, *C. soldanella* and *V. rotundifolia*, but Na2 inhibited the above-ground biomass of *C. soldanella* and *V. rotundifolia* (Table 2 and Fig. 7). *G. littoralis* and *C. soldanella* allocated more biomass to above-ground organs such as the leaf and stem, but *V. rotundifolia* allocated more biomass to above-ground organ or root (Fig. 8). Nitrogen supply caused *G. littoralis*, *C. soldanella* and *V. rotundifolia* to allocate more biomass to under-ground organs, but salt

**Table 2 One-way ANOVA of effects of salt stress and nitrogen supply on seed germination and seedling growth of three halophytes.**

| Plant traits | Source of variation | df | Glehnia littoralis | | | Calystegia soldanella | | | Vitex trifolia | | |
|---|---|---|---|---|---|---|---|---|---|---|---|
| | | | MS | F-value | p-value | MS | F-value | p-value | MS | F-value | p-value |
| Cumulative germination | Na | 2 | 192.93 | 2.18 | 0.14 | 20.80 | 2.55 | 0.10 | 334.86 | 1.43 | 0.27 |
| | N | 2 | 1,1306.69 | 224.73 | <0.05 | 4.33 | 0.58 | 0.57 | 3,012.39 | 18.32 | <0.05 |
| Leaf number | Na | 2 | 0.46 | 5.58 | <0.05 | 0.40 | 0.27 | 0.77 | 0.40 | 0.27 | 0.77 |
| | N | 2 | 2.61 | 0.00 | <0.05 | 6.93 | 4.11 | <0.05 | 6.93 | 4.11 | <0.05 |
| Hight | Na | 2 | 0.12 | 0.17 | 0.85 | 2.92 | 0.97 | 0.39 | 2.92 | 0.97 | 0.39 |
| | N | 2 | 66.50 | 306.06 | <0.05 | 6.69 | 2.44 | 0.11 | 6.69 | 2.44 | 0.11 |
| Root length | Na | 2 | 10.27 | 1.54 | 0.24 | 21.93 | 1.74 | 0.20 | 21.93 | 1.74 | 0.20 |
| | N | 2 | 533.85 | 134.99 | <0.05 | 163.28 | 10.73 | <0.05 | 163.28 | 10.73 | <0.05 |
| Above biomass | Na | 2 | 0.00 | 0.04 | 0.96 | 0.01 | 2.09 | 0.14 | 0.01 | 2.09 | 0.14 |
| | N | 2 | 0.02 | 46.02 | <0.05 | 0.00 | 0.50 | 0.61 | 0.00 | 0.50 | 0.61 |
| Below biomass | Na | 2 | 0.00 | 1.15 | 0.34 | 0.00 | 0.17 | 0.84 | 0.00 | 0.17 | 0.84 |
| | N | 2 | 0.00 | 23.30 | <0.05 | 0.02 | 0.60 | 0.56 | 0.02 | 0.60 | 0.56 |
| Total biomass | Na | 2 | 0.00 | 0.15 | 0.86 | 0.02 | 0.50 | 0.61 | 0.02 | 0.50 | 0.61 |
| | N | 2 | 0.03 | 49.07 | <0.05 | 0.02 | 0.36 | 0.70 | 0.02 | 0.36 | 0.70 |
| Biomass allocation | Na | 2 | 18.78 | 0.551 | 0.585 | 82.933 | 1.839 | 0.191 | 57.064 | 2.598 | 0.093 |
| | N | 2 | 17,662.14 | 441.00 | <0.05 | 2,172.48 | 32.72 | <0.05 | 196.82 | 9.97 | <0.05 |

**Note:**
Na, salt stress; N, nitrogen supply.

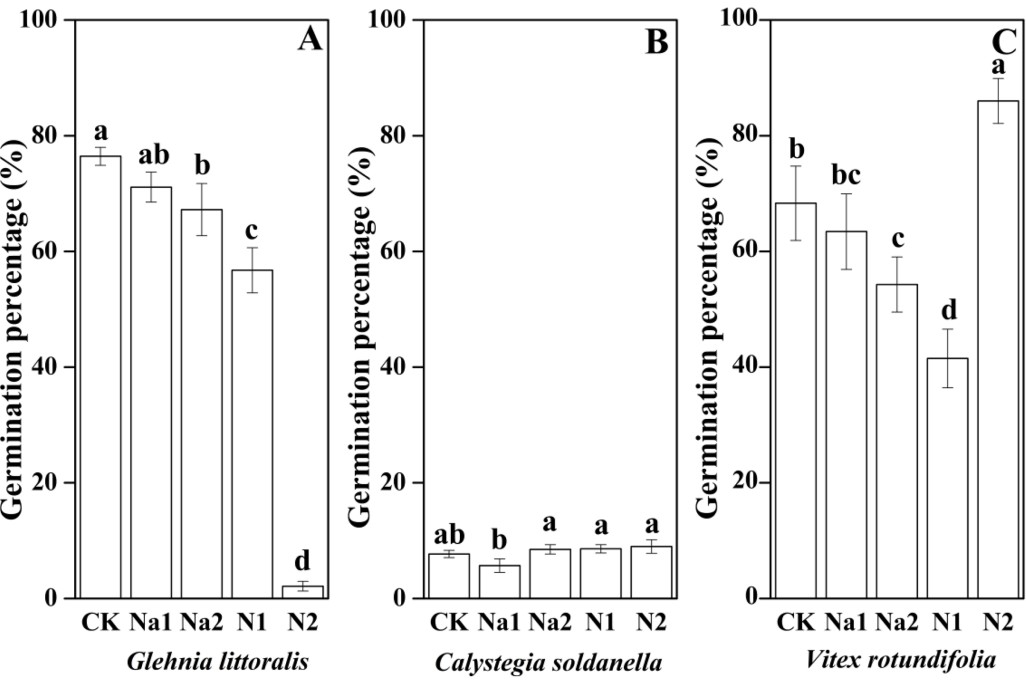

**Figure 3 Effects of salt stress and nitrogen supply on germination percentage of *Glehnia littoralis* (A), *Calystegia soldanella* (B) and *Vitex rotundifolia* (C).** CK, no additional salt and nitrogen were added, Na1, Na$^+$ concentration is about 10 g/kg, Na2, Na$^+$ concentration is about 20 g/kg, N1, 6 gm$^{-2}$year$^{-1}$, N2, 12 gm$^{-2}$year$^{-1}$. Different lowercase letters indicate significant differences ($p < 0.05$) among CK, N1, N2, Na1 and Na2.

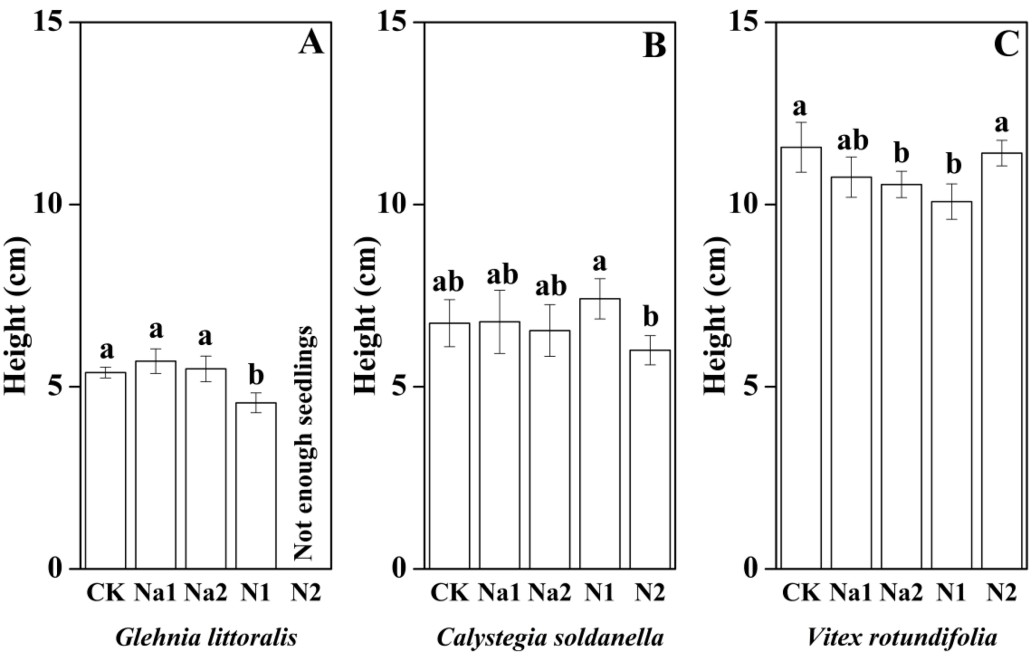

**Figure 4 Effects of salt stress and nitrogen supply on height of *Glehnia littoralis* (A), *Calystegia soldanella* (B) and *Vitex rotundifolia* (C).** CK, no additional salt and nitrogen were added, Na1, Na$^+$concentration is about 10 g/kg, Na2, Na$^+$concentration is about 20 g/kg, N1, 6 gm$^{-2}$year$^{-1}$, N2, 12 gm$^{-2}$year$^{-1}$. Different lowercase letters indicate significant differences ($p < 0.05$) among CK, N1, N2, Na1 and Na2.

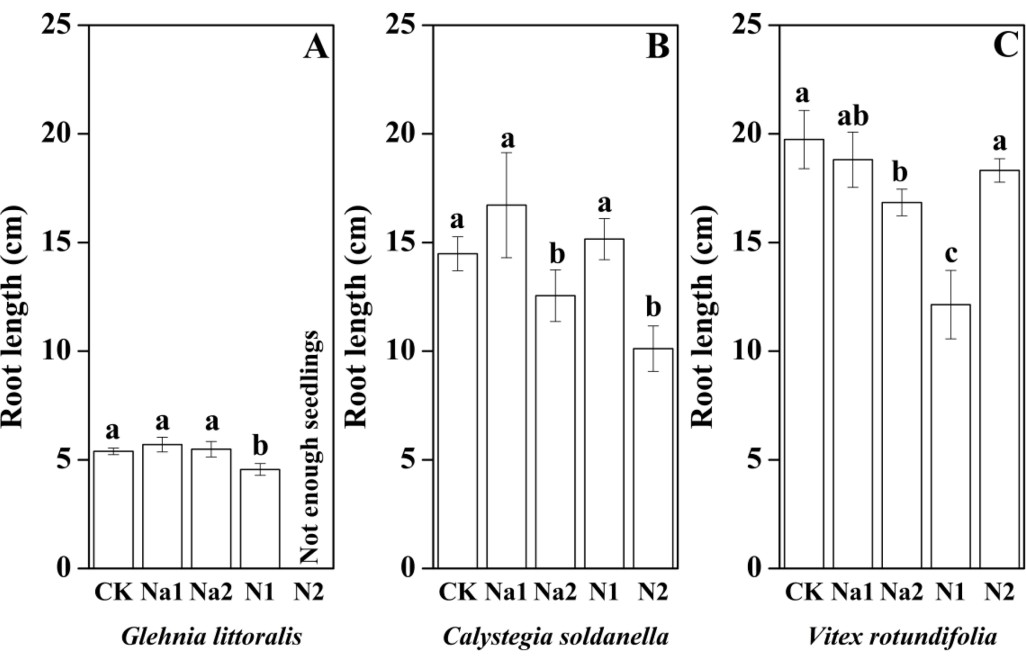

**Figure 5 Effects of salt stress and nitrogen supply on root lenght of *Glehnia littoralis* (A), *Calystegia soldanella* (B) and *Vitex rotundifolia* (C).** CK, no additional salt and nitrogen were added, Na1, Na$^+$concentration is about 10 g/kg, Na2, Na$^+$concentration is about 20 g/kg, N1, 6 gm$^{-2}$year$^{-1}$, N2, 12 gm$^{-2}$year$^{-1}$. Different lowercase letters indicate significant differences ($P < 0.05$) among CK, N1, N2, Na1 and Na2.

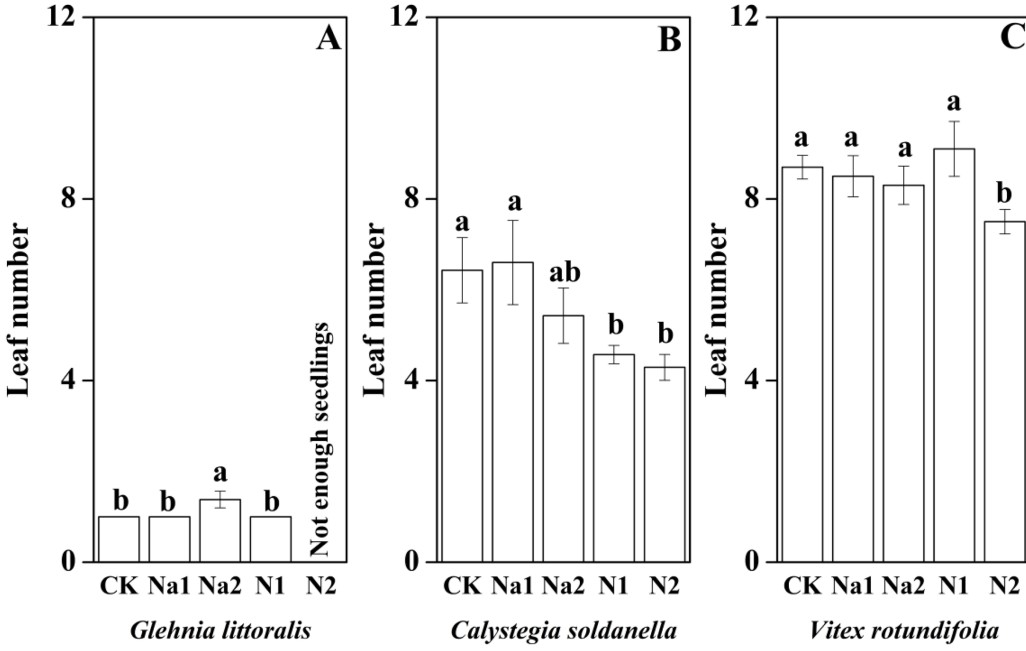

**Figure 6 Effects of salt stress and nitrogen supply on leaf number of *Glehnia littoralis* (A), *Calystegia soldanella* (B) and *Vitex rotundifolia* (C).** CK, no additional salt and nitrogen were added, Na1, Na$^+$concentration is about 10 g/kg, Na2, Na$^+$concentration is about 20 g/kg, N1, 6 gm$^{-2}$year$^{-1}$, N2, 12 gm$^{-2}$year$^{-1}$. Different lowercase letters indicate significant differences ($p < 0.05$) among CK, N1, N2, Na1 and Na2.                                                     

stress had no significant effect on biomass allocation of *G. littoralis*, *C. soldanella* and *V. rotundifolia* (Table 2 and Fig. 8).

## DISCUSSION

This study demonstrated that the three coastal halophytes were more sensitive to nitrogen than salt during the seed germination stage. High nitrogen supply significantly delayed the seed germination and reduced the germination percentage, particularly for *G. littoralis*. Nitrogen inhibited growth more during the seedling growth stage than during the seed germination stage, indicating that our first hypothesis, that the seedling growth stage of halophytes was more sensitive than the seed stage was supported. Nitrogen supply significantly reduced the individual biomass of *G. littoralis*, *C. soldanella* and *V. rotundifolia*, and the individual biomass of the dominant species decreased more significantly than that of the constructive species. *G. littoralis* and *C. soldanella* allocated more biomass to underground organs with increasing nitrogen supply, indicating that the constructive species were less sensitive to nitrogen than the dominant species, and the constructive species exhibited stronger stability than the dominant species. Therefore, our second hypothesis was not support.

Seed germination is the initial stage of the plant life history, which strongly affects seedling establishment, individual development and community stability (*Milla & Lopez, 2014*). Seed germination is often affected by both seed dormancy and external environmental factors (*Jiang et al., 2016*; *DeMalach, Kigel & Sternberg, 2020*; *Zhang et al.,*

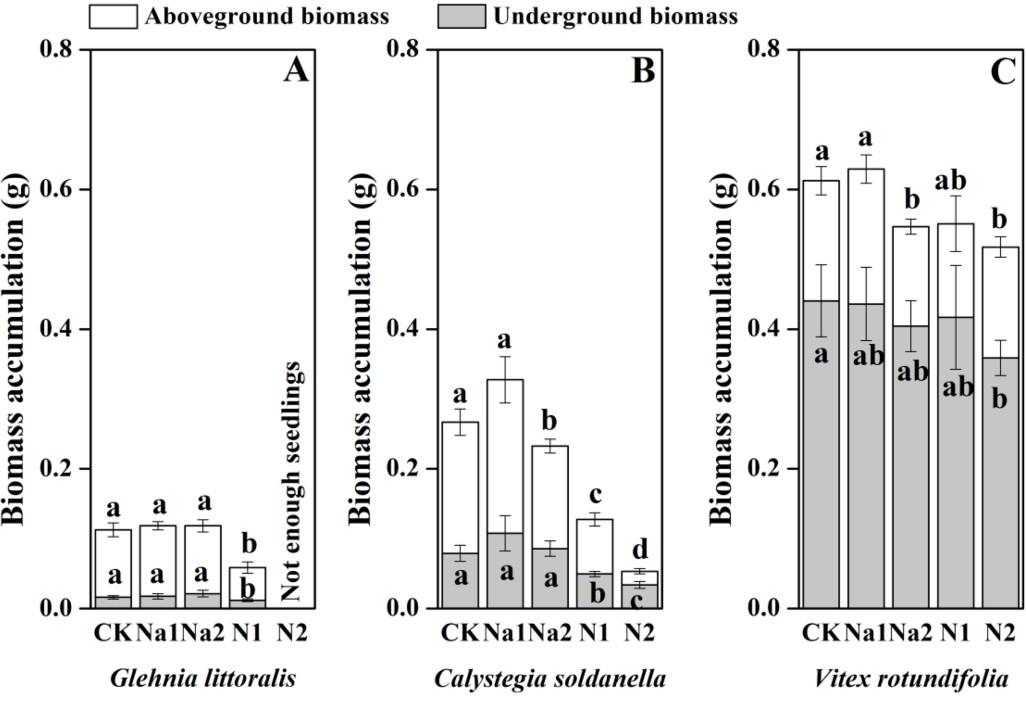

**Figure 7** Effects of salt stress and nitrogen supply on biomass accumulation of *Glehnia littoralis* (A), *Calystegia soldanella* (B) and *Vitex rotundifolia* (C). CK, no additional salt and nitrogen were added, Na1, Na$^+$concentration is about 10 g/kg, Na2, Na$^+$concentration is about 20 g/kg, N1, 6 gm$^{-2}$year$^{-1}$, N2, 12 gm$^{-2}$year$^{-1}$. Different lowercase letters indicate significant differences ($p < 0.05$) among CK, N1, N2, Na1 and Na2.

*2020*). In this study, *G. littoralis* seeds exhibited an explosive germination in the early spring of 2021, which suggested that the seed dormancy of *G. littoralis* has been broken or existed in the form of conditional dormancy. As the temperature gradually rose and the frequency of precipitation increased in the early spring of 2021 (Fig. S2a), the seeds of *G. littoralis* germinated into seedlings quickly. Furthermore, considering the seed dormancy classification system proposed by *Baskin & Baskin (2004)*, seeds of *G. littoralis* have morphophysiological dormancy (*Tada, Kondo & Fuji, 2013*). If seeds of morphophysiological dormancy are buried in the soil after seed dispersal in summer, they will inevitably experience a period of high temperature stratification in the coastal zones, which usually accelerates the development of embryo (*Sharifzadeh & Murdoch, 2001*; *Heydel & Tackenberg, 2017*). After that, the seeds also experienced a period of low temperature stratification (November-March, Fig. S2a), which helped break the physiological dormancy of the seeds and promoted the seed germination with sufficient precipitation under suitable temperature conditions. However, the seed germination percentage did not reach 80%. In our personal observations in field surveys, we found that *G. littoralis* continued to have flowers and young fruit that did not progress from flowering to full fruit maturity. Even in the mature fruit, there were a considerable proportion of empty seeds resulting in the seed germination percentage of less than 80% (Fig. 2). Seed germination percentage of *C. soldanella*, only reached 6% until the spring of the second year, meaning that most seeds of *C. soldanella* had been stored in the soil by seed bank.

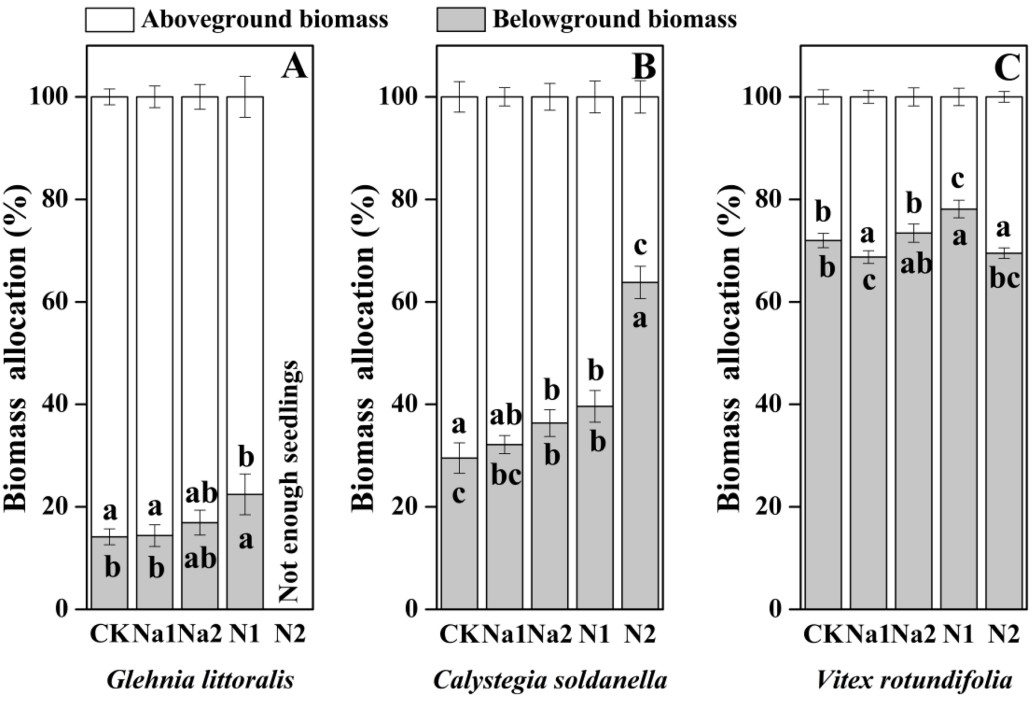

**Figure 8 Effects of salt stress and nitrogen supply on biomass allocation of *Glehnia littoralis* (A), *Calystegia soldanella* (B) and *Vitex rotundifolia* (C).** CK, no additional salt and nitrogen were added, Na1, $Na^+$ concentration is about 10 g/kg, Na2, $Na^+$ concentration is about 20 g/kg, N1, 6 $gm^{-2}year^{-1}$, N2, 12 $gm^{-2}year^{-1}$. Different lowercase letters indicate significant differences ($p < 0.05$) among CK, N1, N2, Na1 and Na2.

Previous experiment results also indicated that seed scarification with 3 h of acid pretreatment with 98% $H_2SO_4$ significantly improved the germination percentage by 70–100% (*Ko et al., 2004*). Thus, *C. soldanella* seeds have physical dormancy and their impermeable coat is responsible for the very low germination percentage from intact seeds. Conversely, *V. rotundifolia* seeds showed slow germination until the late spring of the second year when seed germination percentage reached more than 60% (Fig. 3). We speculated that the slow germination may be related to the slow decomposition of endogenous germination suppression. Similarly, *Wang & Chen (2015)* used concentrated $H_2SO_4$ and cutting the peel to break the mechanical barriers of *V. rotundifolia*, but the seed dormancy was not broken. However, treatment of 500 mg. $L^{-1}$ GA3 made the seeds germinate rapidly, which confirms that endogenous germination suppression is the decisive factor affecting seed germination of *V. rotundifolia*.

  Soil in coastal zones has both high salinity and a deficiency of nutrients. Thus, salt and nitrogen in coastal habitats often affect seed germination and seedling growth by creating an osmotic potential external to the seed preventing water uptake (*Palow & Oberbauer, 2009*; *Wang et al., 2013*). But in our study, salt stress has no significant effect on the seed germination of *G. littoralis*, *C. soldanella* and *V. rotundifolia* seeds, which suggested that the salt stress level used in the study did not reach the tolerance threshold of the three halophytes studied, especially for the pot experiment where high soil moisture conditions may have weakened the negative effect of salt stress (Fig. S2b). Furthermore, halophytes

have several mechanisms to adapt to high salt stress in order to avoid salt damage. For example, the salt tolerance of halophytes is stronger than that of non-halophytes, and long-term growth in a high-saline environment forces halophytes to evolve multiple adaptive mechanisms, such as the fleshiness of stems or leaves, the regionalization of ions in cells or tissues, and the secretion of salt through the salt glands (*Yuan, Leng & Wang, 2016*; *Rozentsvet, Nesterov & Bogdanova, 2017*). Meanwhile, nitrogen is not only the most nutrient element that plants absorb from the soil, but also the main way to improve grassland productivity and crop yield (*Brandt, Seabloom & Cadotte, 2019*; *Hestrin et al., 2021*). In this study, nitrogen supply significantly delayed seed germination and reduced the germination percentage of *G. littoralis* and *V. rotundifolia*, but had no significant effect on the seed germination of *C. soldanella* (Figs. 2 and 3). Excessive nitrogen can decrease soil pH, inhibiting the metabolic reaction of halophytes (*Hong, Gan & Chen, 2019*; *Kimmel et al., 2020*); excessive nitrogen can also change the levels of metal ions, abscisic acid, phytochromes, or seed water absorption, thereby inhibiting seed germination and seedling emergence (*Ochoa-Hueso & Manrique, 2010*; *Yan et al., 2016*). Some studies have confirmed that excessive nitrogen caused a short term drop in productivity and biodiversity loss in a community (*Niu et al., 2008*), so, the delay and decrease in seed germination percentage seen in our study may provide a new explanation for biodiversity loss in the coastal community. Therefore, *C. soldanella* and *V. rotundifolia*'s ability to increase seed germination in response to nitrogen supply may give them a competitive advantage with increased nitrogen deposition.

In the seed germination stage, *V. rotundifolia* germinated later than *C. soldanella* and *G. littoralis*, but the morphological traits and individual biomass of *V. rotundifolia* were larger than that of *C. soldanella* and *G. littoralis*, suggesting that seedlings of *V. rotundifolia* have higher growth efficiency. Because of *V. rotundifolia* is a deciduous shrub species, its stems and roots are highly lignified, which is also an important reason that the individual biomass of *V. rotundifolia* was higher than that of perennial herb *C. soldanella* and *G. littoralis*. Nitrogen is also an essential nutrient element for plant growth (*Santi, Bogusz & Franche, 2013*; *Hestrin et al., 2021*), but it uniformly inhibited the growth of three halophytes in this study, with the seedling growth stage being more sensitive to nitrogen supply than the seed germination stage for all three halophytes. The seeds of all three halophytes in this study generally have thick seed coats (*C. soldanella*) or pericarps (*V. rotundifolia* and *G. littoralis*), which protect the embryo from direct exposure to high salt environments (*Chen et al., 2007*). However, the high concentration of nitrogen impacted the ion balance, changed the pH and accelerated the corrosion of the seed coats or pericarps (*Zhang et al., 2015*). When the seedlings entered the rapid growth period, the high concentration nitrogen also increases the concentration of ferric iron and aluminum in soils through ion exchange, which can be toxic to seedlings and can inhibit seedling growth (*Liu, Zhang & Lal, 2016*). Conversely, the seedling growth of all three halophytes exhibited strong tolerance to salt stress, which differed from the results of a previous study that seedling growth was promoted by a low concentration of salt and inhibited by a high concentration of salt (*Zhang et al., 2018*). In our study, the perennial herbs (*G. littoralis* and *C. soldanella*) allocated more biomass to above-ground organs (leaf and stem), but shrub

(*V. rotundifolia*) allocated more biomass to root, which may be because of the difference in plant type. In general, perennial herbs often reproduce within 3–5 years after seed germination, while shrub need a much longer time, sometimes 8–10 years, in order to reach the reproductive period (*Dechamps et al., 2010*). More biomass allocated to the roots in the seedling stage could effectively enhance the absorption of ground water and nutrients by the root, thereby stimulating the growth of seedlings in the next year. In our study, *G. littoralis*, *C. soldanella* and *V. rotundifolia* allocated more biomass to underground organs under different nitrogen supply, which is different from the result of a previous study that nitrogen promoted the aboveground biomass allocation in grasslands (*Aan et al., 2014*). The reason our results differed was likely related to the growth stage of the plant, as plants demand more water and nutrients for growth in the seedling stage, so the plants allocated more biomass to the underground organs, which is conducive to seedlings establishment and resource competition. Meanwhile, the regularity of biomass allocation at different life history stages also found that the growth rate of plants gradually decreases, and the root-to-shoot ratio decreases accordingly.

## CONCLUSIONS

Our findings show that the three halophytes studied were more sensitive to nitrogen than salt stress during the seed germination stage. A high nitrogen supply significantly delayed seed germination and reduced the germination percentage, particularly for *G. littoralis*, but salt stress did not have negative effect on seed germination of the three halophytes. Nitrogen supply also impacted seedling growth more than seed germination, which also suggested the seedling growth stage of halophytes is more sensitive than the seed stage. High nitrogen significantly reduced the individual biomass of *G. littoralis*, *C. soldanella* and *V. rotundifolia*, and the individual biomass of the dominant species decreased more significantly than that of the constructive species. Conversely, nitrogen supply increased the underground biomass allocation of *G. littoralis* and *C. soldanella*, which suggests that the constructive species in the community are less sensitive to nitrogen and have more stability than the dominant species. Therefore, increasing nitrogen supply in the future may firstly affect the growth of the dominant species, and then cause changes in the community structure of coastal halophytes.

## ACKNOWLEDGEMENTS

We thank Hui Zhang and Hong Qiao for their help in collecting data.

### Funding

This work was funded by the National Natural Sciences Foundation of China (32101262 and 41977039) and the Taishan Scholars Program of Shandong Province (no. tsqn201812097). The funders had no role in study design, data collection and analysis, decision to publish, or preparation of the manuscript.

## Grant Disclosures

The following grant information was disclosed by the authors:
National Natural Sciences Foundation of China: 32101262 and 41977039.
Taishan Scholars Program of Shandong Province: tsqn201812097.

## Competing Interests

The authors declare that they have no competing interests.

## Author Contributions

- Yanfeng Chen conceived and designed the experiments, performed the experiments, analyzed the data, prepared figures and/or tables, authored or reviewed drafts of the article, and approved the final draft.
- Yan Liu performed the experiments, prepared figures and/or tables, and approved the final draft.
- Lan Zhang analyzed the data, prepared figures and/or tables, and approved the final draft.
- Lingwei Zhang conceived and designed the experiments, authored or reviewed drafts of the article, and approved the final draft.
- Nan Wu conceived and designed the experiments, authored or reviewed drafts of the article, and approved the final draft.
- Huiliang Liu conceived and designed the experiments, authored or reviewed drafts of the article, and approved the final draft.

## Data Availability

The raw measurements are available in the Supplemental File.

## Supplemental Information

Supplemental information for this article can be found online at http://dx.doi.org/10.7717/peerj.14164#supplemental-information.

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
