# Peer review of "Effect of salt stress and nitrogen supply on seed germination and early seedling growth of three coastal halophytes"

_PeerJ, doi:10.7717/peerj.14164_

## Round 0.1 · original submission · Major Revisions

Dear authors

Considering the perspective of the reviewers, please review the paper and resubmit it again.

·

Basic reporting

• The study has been properly conducted and carried out;
• Presents a concise and clear review of the problem to be studied;
• Figures and tables are relevant and illustrative;
• Literature is compliant, sufficient, well distributed, balanced and up to date;
• The raw data provided and the supplementary information are useful for a better understanding.

Experimental design

• The work is original for the region and reality in which it took place;
• The scientific question and hypotheses are correctly formulated;
• The material and methods are very well described and are sufficient to be reproduced in another research.

Validity of the findings

Based on what has been presented, this work has demonstrated scientific coherence and objectivity. Its conclusions will certainly be an important contribution to the study of coastal marginal zones. The data (based on the design) are robust and statistically representative. The discussion and the main conclusions respond to and are consistent with the proposed scientific hypotheses

Additional comments

Globally the manuscript is well written and do not present major slips, but for a detailed analysis in each section see “Specific comments by section”. The title clearly reflects the contents and appropriate keywords were provided. Although the abstract should be improved, the results were in general well-presented and the aim of the work is reached. The organization, length and references of the manuscript are correct. Concerning the English language several suggestions could be made but they configure more a writing style of the authors. The written English does not compromise scientific understanding, nor does it call into question the relevance of the proposed theme. Therefore the English does not need further revision, but the verb tense should be uniform (present or past) along the Results and Discussion. I also suggest a last revision of the guidelines for publication in PeerJ.

Specific comments by section
(Please find below a list of comments and suggestions I hope can improve the quality of manuscript)

• Abstract
Lines 4-9: Put the institutional references numbering in superscript;
line 29: Meanwhile, all tree ……..

• Introduction
line 48: Yu et al., is from 2014 not 2013. Review and amend;
line 51: Nasem et al., is the proper name. Should be Badreldin et al. 2015;
line 66: should be Uslu & Gedik, 2019. Use the ampersand, also known as the and sign;
line 97: should be Song & Xing, 2010;
line 113-115: Aren't they already results? It should not come here;
line 119: Yang et al., 2019. Italic.

• Material & Methods
line 151-153: Aren't they already results? It should not come here;
line 155: Wanpingkou Scenic Spot in Rizhao City, already mentioned above. Remove;
line 180: Zhou et al., 2015 not listed in the references. Attention;
line 184: Fu et al., is the proper name. Should be Yandan et al. 2015.

• Material & Methods
line 169: Table 2 not Table 12;
line 321: Should be Wang & Chen, 2015. Use the ampersand, also known as the and sign;
line 333: … halophytes have several mechanisms. Plural. As described below;
line 375: Zang et al., is from 2018 not 2019. Review and amend.

• References
line 422: Is DeMalach N,………… Review and amend;
line 426: Is Yandan F, not Fu Dandan. Review and amend;
line 473: Standardise according to the rest of the bibliography. Badreldin N, ………;
line 499: Song J,……, is from 2009 not 2010. Review and amend;
line 535: Zhang Y T,……, is from 2019 not 2018. Review and amend.

Reviewer 2 ·

Basic reporting

The paper of Chen et al., Effect of salt stress and nitrogen supply on seed germination
and early seedling growth of three coastal halophyte, addresses an important and relevant topic within a global context. However, from the scientific point of view it does not bring any novelty regarding the different strategies used by halophytes to cope with high salinity, and the analysis of seed germination, seedling growth and biomass are rather simple and not enough to elucidate the key processes undeerlying salt tolerance. I would suggest for instance, a deep physiological and metabolic analysis, complemented by cytological studies and at least on of the "Omics" approaches.

Experimental design

The experimental design seems quite complete and well developed.

Validity of the findings

Although the findings are valid, as mentioned above, the approach is rather incomplete to elucidate the salt tolerance strategies used by these plants.

---

## Round 0.2 · Minor Revisions

Dear Dr. Liu,

Thank you for your submission to PeerJ.

I am writing to inform you that your manuscript - Effect of salt stress and nitrogen supply on seed germination and early seedling growth of three coastal halophytes - still needs revision since there are a lot of grammar errors, especially with the use of adverbs. Professional proofreading is required. Please send a revised version for final analysis.

With kind regards,
Fernando Lidon

Academic Editor, PeerJ

---

## Round 0.3 · accepted · Accept

Dear Dr. Liu,

Thank you for your submission to PeerJ.

I am writing to inform you that your manuscript - Effect of salt stress and nitrogen supply on seed germination and early seedling growth of three coastal halophytes - has been Accepted for publication. Congratulations!

Best regards
Flidon